# Co-Administration of High-Dose Nebulized Colistin for *Acinetobacter baumannii* Bacteremic Ventilator-Associated Pneumonia: Impact on Outcomes

**DOI:** 10.3390/antibiotics13020169

**Published:** 2024-02-08

**Authors:** Ioannis Andrianopoulos, Nikolaos Kazakos, Nikolaos Lagos, Theodora Maniatopoulou, Athanasios Papathanasiou, Georgios Papathanakos, Despoina Koulenti, Eleni Toli, Konstantina Gartzonika, Vasilios Koulouras

**Affiliations:** 1Intensive Care Unit, University Hospital of Ioannina, 45500 Ioannina, Greece; jandri0@yahoo.gr (I.A.); nikazakos@yahoo.com (N.K.); lagman75@gmail.com (N.L.); doramaniatopoulou@yahoo.gr (T.M.); thanasis.papathanasiou@gmail.com (A.P.); gppthan@icloud.com (G.P.); eletoli87@gmail.com (E.T.); vpkoulouras@yahoo.gr (V.K.); 2UQ Centre for Clinical Research, Faculty of Medicine, The University of Queensland, Brisbane, QLD 4029, Australia; 3Second Critical Care Department, Attikon University Hospital, Rimini Street, 12462 Athens, Greece; 4Department of Microbiology, University Hospital of Ioannina, 45500 Ioannina, Greece; kgartzon@uoi.gr

**Keywords:** *Acinetobacter baumannii*, bacteremia, ventilator-associated pneumonia, nebulized colistin, mortality, critically ill

## Abstract

*Acinetobacter baumannii* (*A. baumannii*) is a difficult-to-treat (DTR) pathogen that causes ventilator-associated pneumonia (VAP) associated with high mortality. To improve the outcome of DTR A. *Baumannii* VAP, nebulized colistin (NC) was introduced with promising but conflicting results on mortality in earlier studies. Currently, NC is used at a much higher daily dose compared to the past. Nevertheless, there is little evidence on the effect of high-dose NC on the outcomes of *A*. baumannii VAPs, especially in the current era where the percentage of colistin-resistant *A*. *baumannii* strains is rising. We conducted a retrospective study comparing bacteremic *A. baumannii* VAP patients who were treated with and without NC co-administration and were admitted in the Intensive Care Unit of University Hospital of Ioannina from March 2020 to August 2023. Overall, 59 patients (21 and 38 with and without NC coadministration, respectively) were included. Both 28-day and 7-day mortalities were significantly lower in the patient group treated with NC (52.4% vs. 78.9%, *p* 0.034 and 9.5% vs. 47.4%, *p* 0.003, respectively). Patients treated with NC had a higher percentage of sepsis resolution by day 7 (38.1% vs. 13.5%, *p* 0.023) and were more likely to be off vasopressors by day 7 (28.6% vs. 8.1%, *p* 0.039). The addition of NC in the treatment regime of *A*. *baumannii* VAP decreased mortality.

## 1. Introduction

*Acinetobacter baumannii* (*A*. *baumannii*) is a multi-drug-resistant (MDR) pathogen associated with severe infections, in particular, primary bacteremias, pneumonia and urinary tract infections [1,2]. It is an established difficult-to-treat (DTR) pathogen and its prevalence among hospital infections is rising [3]. Therefore, it is now being considered by the WHO as a critical priority to identify novel treatments for [4]. Mortality in such infections is particularly high among patients with bacteremias, pneumonia or sepsis [5,6,7,8]. It appears that critically ill patients that develop *A*. *baumannii* bacteremia due to ventilator-associated pneumonia (VAP) have worse prognosis, especially when they develop septic shock as a complication [9]. And it seems that the emergence of the COVID-19 pandemic has caused a rise in the prevalence of bacterial superinfections from MDR and DTR pathogens [10,11], as well as in the prevalence of VAP as a cause of bacteremia in critically ill patients [12].

Part of the reason for the *A*. *baumannii* VAP-associated high mortality is the fact that *A*. *baumannii* VAP infections are difficult to treat as therapeutic antibiotic choices are rather limited. In addition, penetration of intravenous antibiotics in the inflamed lung tissue is rather poor for most antibiotics, e.g., colistin b-lactams [13,14]. An attempt to overcome the poor penetration is to increase the dose or reduce the time interval between doses [15], though in cases of higher MIC, even that may not be adequate [16,17]. Therefore, to overcome this, especially in cases of pan-drug-resistant (PDR) pathogens, physicians have pursued combinations of antibiotics, although with conflicting results [18,19,20,21,22,23].

Another strategy to overcome the poor lung tissue penetration in cases of VAP is by using nebulized antimicrobials. Nebulized antimicrobials’ theoretical advantage over the intravenous route is that nebulized antibiotics are delivered directly in the lung alveoli, and both experimental and clinical studies have shown that they offer far better lung tissue concentrations of important antibiotics such as aminoglycosides and colistin compared to the intravenous (IV) route [24,25]. Nebulized colistin, in particular, has been the subject of investigation in VAP from MDR pathogens in a few studies, with conflicting results on its effect on patient outcome. This inconsistency in results has been attributed to the differences in study population, dose of nebulized colistin, co-administration of IV colistin or other intravenous antibiotics and the use of different types of nebulizers [26,27]. In the most recently published meta-analysis, nebulized colistin showed a higher eradication rate but failed to show a statistically significant mortality rate improvement [26]. Nevertheless, there was significant heterogeneity among studies, and there is a need for further trials. This necessity for new trials is highlighted first of all by the fact that currently much higher doses (12–15 million units of colistin per day) than those studied in these earlier studies are used, and the effect of these higher doses on patient outcome is understudied [27].

In addition, as the percentage of DTR and colistin-resistant strains is rising, therapeutic choices become very limited, especially in cases of PDR strains or infections where it is hard to achieve adequate antibiotic concentrations, e.g., VAP and meningitis. In such cases, physicians end up using empirically used combinations of antibiotics, while evidence on the best combination remains scarce [28,29]. In PDR *A*. *baumannii* VAP cases, the combination of colistin, tigecycline and high dose sulbactam has showed promising results [30]. Interestingly, other studies also suggest that colistin appears to be the cornerstone of treatment, irrespective of its sensitivity status, especially when cefiderocol is unavailable as is the case in Greece [22]. If such is the case, then colistin is an essential drug in DTR *A*. *baumannii* VAP, and therefore, the hypothesis of improving treatment outcomes using adjunct nebulized colistin in addition to IV is raised.

The purpose of our study was to assess the effect of nebulized colistin in addition to the intravenous antibiotics on the mortality and treatment outcome of bacteremic DTR *A. baumannii* VAP. 

## 2. Results

Overall, from 144 patients with bacteremia from *A*. *baumannii*, we identified 70 (48.6%) patients that had VAP as the source of their bacteremia. After excluding patients that received less than 24 h of antibiotics and patients that developed brain death, our analysis included 59 patients (44 male, 15 female) of mean age of 69 ± 9.6 years old. Out of these 59 patients, 21 received nebulized colistin as part of their antibiotic treatment regime (Group A) and 38 did not receive nebulized colistin as part of their antibiotic treatment regime (Group B). Eighteen (18) patients received 5 million units of nebulized colistin every 8 h, 1 patient received 6 million units of nebulized colistin every 8 h, and 2 patients received 3 million units every 8 h.

Baseline characteristics of the two groups of patients with VAP and associated bacteremia due to *A. baumannii* are shown in Table 1. Overall, there was no statistically significant difference in the baseline characteristics between the subgroup of patients that received combination of IV antibiotics and nebulized colistin (Group A) and the subgroup that received only IV antibiotics (Group B). Patients from Group A and B were of similar age (67.5 ± 9.8 and 67.3 ± 9.6, respectively, ns) and had similar SOFA (3 vs. 3, respectively, ns) and similar APACHE II (19 vs. 20, respectively, ns) scores. There was also no statistically significant difference in comorbidities. Regarding antimicrobial resistance, although Group A patients had a tendency for more resistant strains (a higher percentage of PDR, 28.6% vs. 13.2%, and a higher percentage of colistin resistance, 91.5% vs. 71.1%), altogether the proportion of PDR and colistin-resistant *A*. *baumannii* strains did not statistically differ between the two study groups. The great majority of patients were treated with a combination of antibiotics, while 33.3% of Group A patients and 31.6% of Group B patients received a triple combination of antibiotics (no statistical difference). Finally, there was no statistical difference between the two groups regarding the use of intravenous colistin (90.5% Group A vs. 78.9% in Group B, *p* = 0.258). 

The outcomes of both patient groups are shown in Table 2. Overall, the 28-day mortality was 69.5%. Both all-cause 28-day mortality and 28-day mortality related to *A. baumannii* infection were significantly lower in the group of patients that received nebulized colistin: all-cause 28-day mortality was 52.4% in Group A vs. 78.9% in Group B, *p* 0.034, and 28-day mortality attributed to *A*. *baumannii* was 40% vs. 73.5% in Group A and B, respectively, *p* 0.025) (Figure 1). 

Group A patients receiving nebulized colistin also had a significantly reduced 7-day mortality (9.5% vs. 47.4%, *p* = 0.003), a shorter resolution of sepsis by day 7 (38.1% vs. 13.5%, *p* = 0.023) and were more likely to be without the need for vasopressors by day 7 (28.6% vs. 8.1%, *p* = 0.039). Among the two groups, there was no statistical difference in terms of complications, i.e., acute respiratory distress syndrome, coagulopathy, liver dysfunction and sepsis. Patients that were not treated with nebulized colistin had a tendency to develop more frequent septic shock, though results failed to reach statistical significance (66.7% in Group A vs. 86.8% in Group B patients, *p* = 0.065). Regarding safety considerations, colistin treatment was generally well-tolerated in both patient groups, and only in one patient in Group B, intravenous colistin was discontinued prematurely due to nephrotoxicity; the development of acute kidney injury and the need for renal replacement therapy was similar in both groups (33.3% in Group A vs. 40% in Group B, *p* = 0.587).

Finally, we performed a sensitivity analysis using a propensity score analysis. A propensity score covariate was formed and adjusted for baseline characteristics (age, disease severity and comorbidities), and a multivariate binary logistic regression analysis was performed, that included the propensity score covariate, with the PDR, nebulized colistin and high dose ampicillin/sulbactam used as covariates. The analysis showed that nebulized colistin was an independent risk factor for mortality (Appendix A). 

## 3. Discussion

Our study of 59 critically ill patients with DTR *A. baumannii* VAP and associated bacteremia showed that the co-administration of nebulized colistin in the standard treatment regime was accompanied by a reduction in the 28-day all-cause mortality (52.4% vs. 78.9%, *p* = 0.025). It also showed that the 28-day mortality attributed to DTR *A. baumannii* and the 7-day mortality were significantly lower in the patient group treated with nebulized colistin (40% and 9.5% vs. 73.5% and 47.4%, respectively). An important finding of our study was also that those patients treated with nebulized colistin were more likely to be off vasopressors by day 7 (28.6% vs. 8.1%) and were more likely to have resolution of their sepsis by day 7 (38.1% vs. 13.5%, respectively). Interestingly, the great majority of patients (90.5% Group A, 71.1% Group B) had VAP from colistin-resistant *A*. *baumannii*. Also, as expected by the PK characteristics of nebulized colistin and in line with previous studies, the beneficial effects of nebulized colistin co-administration were not accompanied by an increased incidence of acute kidney injury and need for renal replacement therapy [31,32]. 

To the best of our knowledge, this is the first study that shows improvement in mortality in patients with bacteremic *A. baumannii* VAP receiving nebulized colistin in addition to an intravenous antibiotic regime that, in the vast majority of cases, includes colistin. Two previous studies, the study of Tumbarello et al. and the study of Korbila et al., did show a higher clinical cure rate in the nebulized group but failed to show improvement in mortality [33,34]. A possible explanation for this difference compared to our study might be the low dose of nebulized colistin used in both studies (<3 million units/day) compared to the high dose of 15 million units/day used in our study. Another important difference between our study and the two previous studies was the sensitivity pattern of *A. baumannii*. Specifically, in the studies of Tumbarello et al. and Korbila et al., *A. baumannii* strains were colistin-sensitive, while the majority of the strains in our study were colistin-resistant, and possibly, the addition of nebulized colistin in our cohort, with its associated significant increase in concentration in the epithelial lining fluid, might have made the difference in the overall efficacy of the combined regime. Finally, another factor explaining the difference in mortality outcomes might be the fact that the majority of our patients had COVID-19 pneumonia on ICU admission (90.5% Group A, 89.5% Group B). COVID-19 patients have a diffuse disease of their lung parenchyma and are more vulnerable to superinfections as a significant part of their lung parenchyma is diseased. The addition of nebulized colistin in these patients might significantly enhance the action of colistin compared to non-COVID-19 patients. Unfortunately, the number of non-COVID-19 patients with bacteremic VAP was very low (10%), and no comparison could have been made between COVID-19 and non-COVID-19 patients with bacteremic VAP.

Our study showed that patients receiving nebulized colistin also demonstrated improved 7-day outcomes, in terms of mortality as well as sepsis resolution and weaning off the vasopressors. This is in accordance with most previous studies that have showed better clinical response in patients receiving nebulized colistin [33,34,35,36]. The improvement in 7-day mortality better reflects the outcome of the infection as it is closer to the onset of the infection compared to day 28 [12].

Finally, to the best of our knowledge, our study is the largest one reporting results on the effect of nebulized colistin in colistin-resistant *A. baumannii* VAP, while most studies usually exclude patients with colistin-resistant *A. baumannii* [35]. There are only two other studies addressing this issue in PDR *A. baumannii* patients with VAP: the first is a case series study of 10 patients, who received nebulized and intravenous colistin, high-dose tigecycline and high-dose ampicillin/sulbactam, which showed a very good clinical outcome; and the second one recruited 32 patients with PDR *A. baumannii* VAP: 10 patients were treated with the above mentioned triple antibiotic combination (80% also received fosfomycin) and 22 patients received a combination of two antibiotics (details were not mentioned) [30,37]. The latter study also reported a favorable outcome with improved survival. Therefore, all three studies show a favorable outcome in patients with colistin-resistant VAP treated with nebulized colistin. As colistin resistance of *A. baumannii* strains continues to rise and new antibiotics such as cefiderocol are not yet widely available across countries (as it was also not available in our country during the study period), more studies addressing the effect of colistin nebulization in colistin-resistant *A. baumannii* VAP are urgently needed. 

### 3.1. Limitations

Our study has limitations. First of all, as a retrospective single center study, it is subject to the inherent limitations of such studies. Our study sample is not large; however, it is similar to most studies published on the same subject [27]. Also, the majority of our cohort were COVID-19 patients, and one might argue that more data from non-COVID-19 patients are needed before we extrapolate our conclusions to a non-COVID-19 ICU population. Nevertheless, there is currently no evidence that COVID-19 patients respond differently to nebulized antibiotics. In addition, the *A. baumannii* strains of our cohort were mostly colistin-resistant, and our results might not be generalizable in different settings. Also, there was no record in patient data about which nebulizer was used in each particular patient (nebulized colistin was delivered either with vibrating mesh or jet nebulizer). Nevertheless, despite the fact that, in most cases, a jet nebulizer was used, which is known to provide nebulization in the least optimal way, [38] our study had a positive outcome for nebulized colistin. Finally, as cefiderocol was not available in our country during the study period, our results may not be generalizable in patients receiving cefiderocol as part of their therapeutic regime; nevertheless, results are conflicting regarding cefiderocol’s superiority over colistin for such infections [39]. 

### 3.2. Strengths

To the best of our knowledge, this is the first study comparing patients receiving either a combination of both intravenous and high-dose nebulized colistin or intravenous colistin only. It is also the first study showing improvement in mortality outcomes. Moreover, in most previous studies, nebulized colistin was administered in critically ill patients with gram negative infections attributed to colistin-sensitive *A. baumannii*, while in our study, most patients had colistin-resistant *A. baumannii*; nearly 30% of Group A patients had PDR *A. baumannii*. The presence of a significant percentage of colistin-resistant and PDR *A. baumannii* strains reflects the current (unfortunate) reality in a significant number of ICUs in several countries globally as it has been shaped since the COVID-19 pandemic. The emergence of colistin resistance or failure in particular could be related to subtherapeutic levels of colistin in patients, underlining perhaps the importance of using nebulized colistin in VAP [40]. 

## 4. Materials and Methods

We conducted a retrospective, single center, observational study that included all patients with bacteremic VAP due to *A*. *baumannii* who were admitted in the two intensive care units (ICU) of Ioannina University Hospital from March 2020 to August 2023. University Hospital of Ioannina is a 900-bed academic hospital in Ioannina, Greece. Data were collected in an anonymized way, and data collection was approved by the institutional review board of the University Hospital of Ioannina (protocol number 37/2023). Due to the retrospective, observational nature of the study, the need for informed consent was waived.

### 4.1. Data Collection and Patient Groups

We recorded information on comorbidities, sex, age, severity of disease, including severity of disease scores on admission such as the Sequential Organ Failure Assessment (SOFA) and the Acute Physiologic Assessment and Chronic Health Evaluation (APACHE) II. We also recorded the SOFA score the previous day of sepsis onset, on the day of sepsis development and on day 7 since the development of sepsis. In addition, we recorded complications, including acute respiratory distress syndrome (ARDS), acute kidney injury (AKI), the application of continuous renal replacement therapy (CRRT), coagulopathy, sepsis, septic shock and hepatic dysfunction. We also recorded 7-day outcomes such as resolution of sepsis, discontinuation of vasopressors and microbiological cure, and long term outcomes such as 28-day mortality, association of death with the *A. baumannii* infection, antibiotic sensitivities and the type, dose and duration of antibiotics administered, ICU length of stay (LOS) and duration of mechanical ventilation. Patients that did not receive antibiotics with potential action against *A. baumannii* or received antibiotics for less than 24 h were excluded from our study. In patients with more than one episode of bacteremia from *A. baumannii*, only the first episode was recorded.

Patients were categorized into two groups: in Group A were patients with VAP and a secondary *A. baumannii* bacteremia that received nebulized colistin as part of their antibiotic treatment regime, and in Group B were patients with VAP and a secondary *A. baumannii* bacteremia that did not receive nebulized colistin as part of their antibiotic treatment regime.

### 4.2. Study Outcomes

The primary outcomes of the study were 28-day total mortality and 28-day mortality related to the *A. baumannii* infection. Secondary outcomes included the development of sepsis, the 7-day mortality, the 7-day resolution of sepsis, the 7-day need for vasopressors, the 7-day microbiological cure, the length of stay in the ICU and in the hospital, the duration of mechanical ventilation and the development of complications-organ failures.

### 4.3. Definitions

We defined ventilator-associated pneumonia (VAP) by the Centers for Disease Control and Prevention (CDC) criteria [41]. SEPSIS-3 definition was used to define sepsis and septic shock [42]. The Berlin definition was used to define ARDS [43]. For patients already diagnosed with ARDS on admission, both a worsening of the severity of the ARDS stage and a deterioration of the lung opacities were considered as new ARDS complicating the *A*. *baumannii* VAP [43]. We defined acute kidney injury (AKI) based on the ‘Kidney Disease: Improving Global Outcomes’ (KDIGO) guidelines [44]. Hepatic dysfunction was defined, as previously described, as total bilirubin greater than 2 mg/dL [12]. Coagulopathy was defined as, either a platelet count of less than 150,000/μL, or an International Normalized Ratio (INR) of more than 1.4 [45]. Death was attributed to *A. baumannii* if at least two ICU physicians concluded that the patient died from the *A. baumannii*-studied VAP. A SOFA score on day 7 from sepsis less or equal to the 24 h before sepsis SOFA score was defined as resolution of sepsis [41]. Microbiological cure was defined as a negative blood culture result for *A*. *baumannii* in a subsequent blood culture taken up to day 7 post the episode of bacteremia. The European Committee on Antimicrobial Susceptibility Testing (EUCAST) minimum inhibitory capacity (MIC) breakpoints were used to determine sensitivity of antibiotics [46]. MIC for colistin and tigecycline were determined by using broth microdilution, while for the rest of antibiotics the VITEK© system was used. DTR was defined as *A*. *baumannii* resistant to all first class antibiotics (beta-lactams, including carbapenems and fluoroquinolones [47].

### 4.4. Statistical Analysis

Statistical analysis was performed as previously described [12]. The Kolgomorov–Smirnov test was used to assess for normal distribution of continuous variables, and variables were presented as either median, min–max or mean ± standard deviation values as appropriate. The study groups were compared for the continuous variables either by the *t*-test or Mann–Whitney test for the continuous variables whereas, for categorical variables, χ^2^ tests (*p*-values were extracted from Fisher’s exact test) were applied. The log-rank test was used to construct and compare the Kaplan–Meier survival curves of the two groups, after the proportionality of hazards was assessed graphically and confirmed for all covariates. A *p*-value of less than 0.05 was used to define statistical significance in all cases. A sensitivity analysis was performed using propensity score analysis. A propensity score was adjusted for baseline characteristics (age, severity of disease, comorbidities) and used as covariate in binary logistic regression analysis to assess for risk factors of mortality. SPSS^®^ 29.0 was used for all statistical analyses.

## 5. Conclusions

The addition of high-dose nebulized colistin to the standard intravenous treatment of critically ill patients with bacteremic *A. baumannii* VAP seems to improve patients’ outcome. The co-administration of high-dose nebulized colistin appears to be both beneficial and safe irrespective of the in vitro sensitivity of colistin against *A. baumannii*. 

## Figures and Tables

**Figure 1 antibiotics-13-00169-f001:**
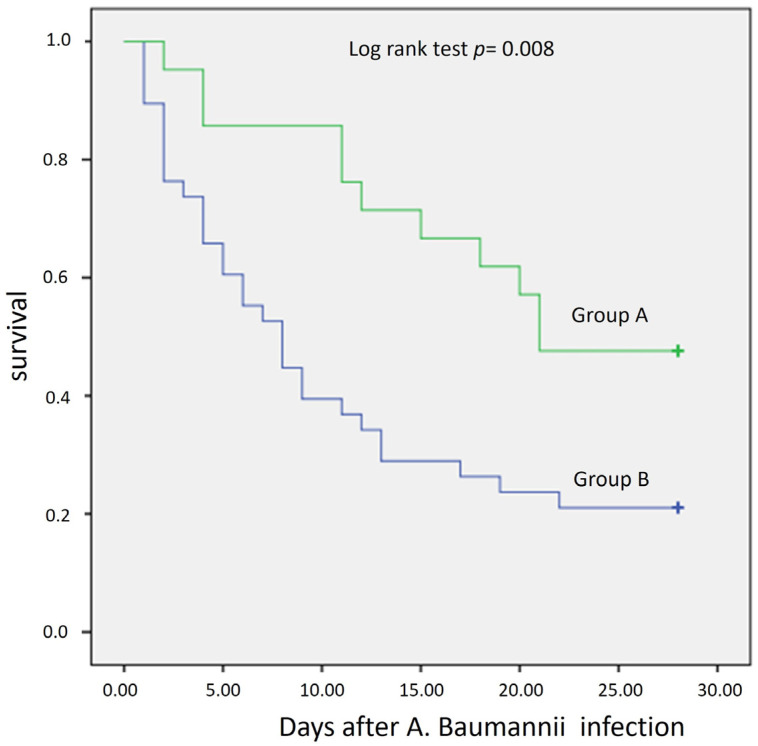
Probability of survival for the first 28 days post *Acinetobacter baumannii* bacteremic VAP was significantly higher for patients receiving nebulized colistin (Group A) vs. patients not receiving nebulized colistin as part of their therapeutic regime.

**Table 1 antibiotics-13-00169-t001:** Baseline characteristics of the patients with bacteremic VAP from *Acinetobacter baumannii* that were treated (Group A) and not treated (Group B) with nebulized colistin.

Parameter	Group A (Treated with Nebulized Colistin) (*n* = 21)	Group B (Treated without Nebulized) Colistin (*n* = 38)	*p*-Value
Age (mean ± SD)	67.5 ± 9.8	67.3 ± 9.6	0.657
Gender (male *n*, %)	15 (71.4%)	29 (76.3%)	0.680
APACHE II median (min–max)	19 (12–35)	20 (12–47)	0.934
CCI ^∞^ median (min–max)	3 (1–9)	3 (0–8)	0.898
SOFA ^∞^ score median (min–max)	4 (3–11)	4 (2–17)	0.798
COVID 19 (*n*, %)	19 (90.5%)	34 (89.5%)	0.903
Comorbidities			
Diabetes mellitus (*n*, %)	10 (47.6%)	14 (36.8%)	0.420
Heart failure (*n*, %)	1 (4.8%)	2 (5.3%)	0.933
Coronary Heart Disease (*n*, %)	3 (14.3%)	3 (7.9%)	0.437
Chronic Kidney Disease (*n*, %)	1 (4.8%)	3 (7.9%)	0.647
Cirrhosis (*n*, %)	1 (4.8%)	0 (0%)	0.175
COPD ^∞^ (*n*, %)	3 (14.3%)	5 (13.2%)	0.904
Obesity (*n*, %)	4 (19%)	5 (13.2%)	0.547
Cancer (*n*, %)	0(0%)	(4 10.5%)	0.124
Infection parameters			
PDR ^∞^	6 (28.6%)	5 (13.2%)	0.146
Colistin sensitive	2 (9.5%)	11 (28.9%)	0.085
Colistin treated	19 (90.5%)	30 (78.9%)	0.258
Patient was treated with:			
Monotherapy	0 (0)	2 (5.3%)	0.443
2 drugs	10 (47.6%)	21 (55.3%)	
3 drugs	7 (33.3%)	12 (31.6%)	
4 drugs	4 (19%)	3 (7.9%)	
Antibiotics			
Tigecycline (*n*, %)	21 (100%)	37 (97.4%)-	1
High dose ampicillin/sulbactam (*n*, %)	13 (62%)	8 (28.1%)	0.064
Meropenem (*n*, %)	7 (33.3%)	15 (39.5%)	0.781
Fosfomycin (*n*, %)	3 (14.3%)	4 (10.5%)	0.69
Appropriate antibiotic treatment in shock ^∫^	17 (85%)	35 (97.2%)	0.089

^∞^ CCI: Charlson Comorbidity Index; SOFA, Sequential Organ Failure Assessment score; COPD, Chronic Obstructive Pulmonary Disease; PDR, pan-drug-resistant. ^∫^ The patient at the time of developing septic shock was already on or was started on antibiotics targeting *A. baumannii.*

**Table 2 antibiotics-13-00169-t002:** Outcome and complications of the patients with bacteremic VAP from *Acinetobacter baumannii* that were treated (Group A) and not treated (Group B) with nebulized colistin.

Parameter	Group A (Treated with Nebulized Colistin) (*n* = 21)	Group B (Treated without Nebulized) Colistin (*n* = 38)	*p*-Value
Mortality			
Total Mortality at 28 days * (*n*, %)	11 (52.4%)	30 (78.9%)	0.034
Mortality at 28 days related to infection * (*n*, %)	6 (40%)	25 (73.5%)	0.025
Mortality at 7 days * (*n*, %)	2 (9.5%)	18 (47.4%)	0.003
Days on MV ^∞^ median (min–max)	28 (11–56)	17 (1–115))	0.053
Ventilator free days @28 d (*n*, %)	4 (19%)	2 (7.8%)	0.465
Median min–max	9 (4–18)	18.5 (17–20)	0.587
ICU days median (min–max)	30 (11–106)	17 (1–144)	0.007
Morbidity related to *A*. *baumannii* infection (*n*, %)	8 (38.1%)	8 (21.1%)	0.159
Free of symptoms at day 7 *	9 (45%)	10 (26.3%)	0.233
Free of vasopressors at day 7 *	6 (28.6%)	3 (8.1%)	0.039
Resolution of sepsis at day 7 *	8 (38.1%)	5 (13.5%)	0.023
Microbial cure at day 7 *	9 (45%)	10 (26.3%)	0.150
Complications related to *A*. *baumanii* infection (*n*, %)			
Sepsis	19 (90.5%)	36 (94.7%)	0.533
Shock	14 (66.7%)	33 (86.8%)	0.065
Cardiomyopathy ^∫^	1 (4.8%)	3 (8.1%)	0.629
AKI ^∞,∫^	7 (33.3%)	15 (40.5%)	0.587
Coagulopathy ^∫^	13 (61.9%)	29 (76.3%)	0.242
Hepatic dysfunction ^∫^	11 (52.4%)	19 (50%)	0.861
ARDS ^∞,∫^	18 (85.7%)	36 (94.7%)	0.233
CRRT ^∞,∫^	2 (9.5%)	4 (10.8%)	0.877

^∞^ Abbreviations: ARDS, Acute Respiratory Distress Syndrome; AKI, Acute Kidney Injury; CRRT, Continuous Renal Replacement Therapy; MV, mechanical ventilation. ^∫^ Coagulopathy was defined as a platelet count < 150 × 10^3^/μL or an INR > 1.4. Cardiomyopathy was defined as new onset left ventricular systolic dysfunction during the septic episode. Hepatic dysfunction was considered when total bilirubin was >2 mg/dL. AKI was defined by the KDIGO criteria. * Total mortality at 7 days and total mortality at 28 days were defined as death occurring between the day of a positive blood culture for *A. baumannii* up until 7 or 28 days, respectively. Mortality was attributed to *A*. *baumannii* if at least two intensive care physicians decided that the patient died from septic shock caused by the *A*. *baumannii* infection. Free of symptoms by day 7 was defined as resolution of symptoms and improvement of oxygenation by day 7 post the *A*. *baumannii* positive blood culture, free of vasopressors by day 7 was defined as the patient not receiving any vasopressors on the seventh day post the bacteremia. Microbiological cure by day 7 was defined as *A. baumannii* not growing in follow-up blood cultures by day 7. Resolution of sepsis was defined as a SOFA score on day 7 ≤ 24 h before the sepsis-onset SOFA plus maximum one more point.

## Data Availability

The datasets used and/or analyzed during the current study are available from the corresponding author on reasonable request.

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
