# Peer review of "Co-Administration of High-Dose Nebulized Colistin for Acinetobacter baumannii Bacteremic Ventilator-Associated Pneumonia: Impact on Outcomes"

_antibiotics, 2024, doi:10.3390/antibiotics13020169_

Round 1

Reviewer 1 Report

Comments and Suggestions for Authors

In their paper entitled: "Co-administration of high-dose nebulized colistin for Acinetobacter baumannii bacteremic ventilator-associated-pneumonia: impact on outcomes", the authors discussed improvement of clinical outcomes. Difficult to treat pathogens causing severe associated-pneumonia with high mortality is being a significant public health issue worldwide. The study discussed here highlights significant points.

The paper is well designed, structured and presented. The readership can easily follow the idea discussed throughout the paper. However, there are still some points to raised and that need to be corrected to completely showcase the soundness of this paper.

Abstract: Although this section is concise and summarizes the significant details of the paper, it really lacks the time frame of the study.

Throughout the text: (1) There couple of sentences that are really quite long and might need to be reformatted. (2)There some quite typos errors that need to be addressed.     

Comments on the Quality of English Language

The English language is readable, fluent although the length of some sentences alters or complicate to sometimes grasp the deep sense of sentences. Moreover, some typos errors should be carefully checked and addressed 

Author Response

Dear reviewer, thank you very much for reviewing our paper and for your kind comments.

Hereby follows a point- by- point answer (in bold) to your comments regarding the revision of our paper.

Abstract: Although this section is concise and summarizes the significant details of the paper, it really lacks the time frame of the study.

 Response: The time frame of the study was added in the abstract.

Throughout the text: (1) There couple of sentences that are really quite long and might need to be reformatted. (2)There some quite typos errors that need to be addressed.

Response: Corrections were made throughout the paper:  typos were corrected and long sentences were reformatted in order for the meaning to be clearer.

Reviewer 2 Report

Comments and Suggestions for Authors

Acinetobacter infections pose a challenge in hospital settings, and few antibiotics are available for treatment. The authors wanted to evaluate the efficacy of aerosolized colistin in a rather limited cohort of patients with a retrospective study design. The study, in my opinion, is very limited by the numbers and by the absence of a propensity score matching that can correct the treatment allocation bias. Acinetobacter infections pose a challenge in hospital settings, and few antibiotics are available for treatment.

The authors want to evaluate the efficacy of aerosolized colistin in a rather limited cohort of patients with a retrospective study design. The study, in my opinion, is very limited by the numbers and by the absence of a propensity score matching that can correct the treatment allocation bias. Secondly, in no case cefiderocol was used, a very promising molecule in Acinetobacter infections today, although the data on its efficacy versus colistin-based regimens (also including aerosolized colistin) are conflicting.

Please if possible consider the suggested statistical methodology and discuss the highlighted points (please see doi: 10.3390/microorganisms11040984, doi: 10.1089/mdr.2021.0109, and doi: 10.1016/j.micpath.2020.104058)

Comments on the Quality of English Language

Minor editing

Author Response

Dear reviewer, thank you very much for reviewing our paper and for your constructive comments.

We agree that study population is small as we pointed out in our limitations, however most studies on this field have similar size population and ours is the only one that addresses bacteremic VAPs with a significant proportion of patients having a colistin resistant strain. Based on your suggestion we performed a propensity score analysis adjusting for patients age, baseline characteristics and comorbidities,and  it was found that nebulized colistin was an independent protective factor for mortality. Results of the propensity analysis are presented in the results section and a supplemental table is added ,as well.

Please if possible consider the suggested statistical methodology and discuss the highlighted points (please see doi: 10.3390/microorganisms11040984, doi: 10.1089/mdr.2021.0109, and doi: 10.1016/j.micpath.2020.104058) Thank you for your useful suggestion.

Response: We read the above references and added information in our manuscript and added them in the reference list.

Reviewer 3 Report

Comments and Suggestions for Authors

In this manuscript, authors undertook an investigation to assess the efficacy of the administration of nebulized colistin along with IV antibiotics in patients suffering from  VAP bacteremias caused by DTR A. baumannii.

A. baumannii needs to be italicized through out the manuscript: Line 35, 40, 77, 102, 225 etc. Please check thoroughly.

Please change the coma to decimal points in %: Line 84, table 2: column 3, row 3

Line 39: Omit "high" before among patients

Line 87-89: Please rephrase. It does not clearly mention the difference of regimens investigated. 

Line 101: Where are the findings of colistin resistance percentages mentioned in Table1? (91.5% vs 71.1%)

Line 116-120: In the biggest observation of reduced mortality in nebulized colistin group, how would you factor in the much higher % of high dose Ampicillin/sulbactam being administered to patients. Is that not a parameter that is significantly different between the two groups examined. Please address.

Line 234: "is" missing before the first study

Comments on the Quality of English Language

English language is OK, only minor edits required.

Author Response

Dear reviewer, thank you very much for your review and your comments. Hereby follows a point- to- point response (in bold) to your comments.

  1. baumannii needs to be italicized through out the manuscript: Line 35, 40, 77, 102, 225 etc. Please check thoroughly.

Response: We italicized A. baumannii throughout the manuscript.

Please change the coma to decimal points in %: Line 84, table 2: column 3, row 3

Response: It is now corrected

Line 39: Omit "high" before among patients:

Response: It is now corrected

Line 87-89: Please rephrase. It does not clearly mention the difference of regimens investigated.  

Response: We added “ as part of their antibiotic treatment regime” to clarify differences among the two groups.

Line 101: Where are the findings of colistin resistance percentages mentioned in Table1? (91.5% vs 71.1%)

Response:  In table 1 in subsection of infection parameters colistin sensitivity is listed. In Group A colistin sensitive strains were 9.5% and in Group B colistin sensitive strains were 28.9% we reversed the percentages in the text to emphasize on the resistance of strains. If reviewers do not think it is appropriate we will make amendments to the main text.

Line 116-120: In the biggest observation of reduced mortality in nebulized colistin group, how would you factor in the much higher % of high dose Ampicillin/sulbactam being administered to patients. Is that not a parameter that is significantly different between the two groups examined. Please address.

Response: While the difference is noticeable the statistical analysis failed to reach statistical significance. In addition, we performed a propensity score adjusted multivariate logistic regression analysis that showed that only nebulized colistin was an independent factor for mortality and not the high dose of ampicillin sulbactam. Findings are presented in the result section.

Line 234: "is" missing before the first study

Response: it is now corrected